# Gene Ontology Analysis Highlights Biological Processes Influencing Non-Response to Anti-TNF Therapy in Rheumatoid Arthritis

**DOI:** 10.3390/biomedicines10081808

**Published:** 2022-07-27

**Authors:** Gregor Jezernik, Mario Gorenjak, Uroš Potočnik

**Affiliations:** 1Faculty of Medicine, University of Maribor, Taborska ulica 8, 2000 Maribor, Slovenia; gregor.jezernik1@um.si (G.J.); mario.gorenjak@um.si (M.G.); 2Faculty of Chemistry and Chemical Engineering, University of Maribor, Smetanova ulica 17, 2000 Maribor, Slovenia; 3Department for Science and Research, University Medical Centre Maribor, Ljubljanska ulica 5, 2000 Maribor, Slovenia

**Keywords:** gene ontology, rheumatoid arthritis, treatment outcome, infliximab, adalimumab, biomarkers

## Abstract

Anti-TNF therapy has significantly improved disease control in rheumatoid arthritis, but a fraction of rheumatoid arthritis patients do not respond to anti-TNF therapy or lose response over time. Moreover, the mechanisms underlying non-response to anti-TNF therapy remain largely unknown. To date, many single biomarkers of response to anti-TNF therapy have been published but they have not yet been analyzed as a system of interacting nodes. The aim of our study is to systematically elucidate the biological processes underlying non-response to anti-TNF therapy in rheumatoid arthritis using the gene ontologies of previously published predictive biomarkers. Gene networks were constructed based on published biomarkers and then enriched gene ontology terms were elucidated in subgroups using gene ontology software tools. Our results highlight the novel role of proteasome-mediated protein catabolic processes (*p* = 2.91 × 10^−15^) and plasma lipoproteins (*p* = 4.55 × 10^−11^) in anti-TNF therapy response. The results of our gene ontology analysis help elucidate the biological processes underlying non-response to anti-TNF therapy in rheumatoid arthritis and encourage further study of the highlighted processes.

## 1. Introduction

Rheumatoid arthritis (RA) is a common complex autoimmune disease characterized by chronic and progressive joint inflammation. Currently, first-line therapeutic approaches in rheumatoid arthritis focus on minimizing disease activity using, primarily, corticosteroids with or without disease-modifying antirheumatic drugs (DMARDs). The development of biological drugs such as monoclonal antibodies against key inflammatory cytokines has significantly improved symptom control [1] in severe rheumatoid arthritis and chronic patients failing first-line therapy. Etanercept [2] and infliximab, inhibitors of proinflammatory cytokine tumor necrosis factor alpha (anti-TNF) [3], were the first anti-TNF biological drugs indicated for rheumatoid arthritis, and later more biological drugs against TNFα were developed, including adalimumab [4], certulizumab pegol [5] and golimumab [6]. In recent years, the emergence of biosimilars of anti-TNF biological drugs has also somewhat reduced the initially high cost of anti-TNF therapy while maintaining efficacy levels comparable to those of the originator biological drugs [7].

However, despite the immense therapeutic power of anti-TNF therapy, 10–30% of patients do not respond to anti-TNF biological drugs upon therapy initiation (i.e., primary non-response) and 23–46% of responders lose response to anti-TNF therapy over time (i.e., secondary non-response) [8]. Non-response to anti-TNF therapy usually represents loss of disease control in patients with severe rheumatoid arthritis, as well as unnecessary exposure to potentially severe adverse effects of anti-TNF drugs and inefficient use of expansive biological therapeutics. Patients who fail to respond to anti-TNF drugs may switch to a different biological drug, such as anakinra, rituximab or sarilumab [9]. Even so, other biological drugs face similar challenges to anti-TNF drugs in terms of non-response [1,10,11]. Therefore, disease-modifying antirheumatic drugs (DMARDs) remain the long-term therapy of choice alongside corticosteroids for disease flares, both of which are known to have significant long-term adverse effects [12].

Predicting non-response to anti-TNF therapy based on the patient’s clinical and biological data would allow targeted therapy with higher efficacy and fewer adverse effects, as well as cost-efficient use of therapeutics. Physicians could determine if and when to switch anti-TNF therapeutics or whether it would be more effective to switch to biological drugs with different therapeutic targets. To date, response to anti-TNF therapy has been intensively studied and several DNA, RNA and protein response biomarkers with low to moderate predictive accuracy have been identified. However, despite the many published anti-TNF response biomarkers, the biological processes underlying non-response to anti-TNF therapy in RA remain largely unknown. Improving the understanding of the mechanisms underlying non-response to anti-TNF drugs on a molecular level would allow the development of novel therapeutic strategies to prevent non-response or the discovery of novel pharmaceutical targets for drug development. To this end, we reviewed already published genomic, transcriptomic and proteomic markers of response and non-response to anti-TNF biological drugs in rheumatoid arthritis and performed a gene ontology analysis to help elucidate biological processes linked to response and non-response to anti-TNF therapy.

## 2. Materials and Methods

### 2.1. Literature Search

To perform a comprehensive review of the literature on anti-TNF therapy response biomarkers, we searched the PubMed database using a combination of terms defining disease, drug, response, biomarker type and exclusion criteria. To prevent Mesh terms missing synonyms, we employed a combination of both Mesh terms and equivalent non-Mesh keywords. The final search query was defined as a combination of the following term groups:Disease terms: “Arthritis, Rheumatoid” (Mesh) OR (“rheumatoid” AND “arthritis”);Drug terms: “infliximab” OR “adalimumab” OR “etanercept” OR “golimumab” OR “certolizumab pegol” OR “Tumor Necrosis Factor-alpha/antagonists and inhibitors” (Mesh) OR “TNFA inhibitor” OR “TNF inhibitor” OR “anti-TNF therapy” OR “anti-TNFA therapy” OR “Treatment Outcome” (Mesh);Response terms: “predictor” OR “responder” OR “nonresponder” OR “non-responder” OR “therapy outcome” OR “therapy response” OR “response biomarker” OR “outcome biomarker” OR “response predictor” OR “outcome predictor”;Biomarker terms: genetics OR genomics OR transcriptomics OR proteomics OR metabolomics OR “DNA methylation”;Exclusion terms: NOT (“tocilizumab” OR dose OR dosing).

Studies were included based on the following inclusion criteria:
Published between the years 2002 and 2022;The study used well-defined response criteria (e.g., those included in the Disease Activity Score in 28 Joints, also known as ΔDAS28);Biomarkers were analyzed prior to therapy initiation and, if applicable, after therapy (e.g., gene expression and serum protein levels);Quantitative biomarkers were reported with a clearly defined direction of association (e.g., gene expression defined as up-regulated or down-regulated, not merely “associated”).

In this gene ontology study, we did not make any additional distinctions based on the anti-TNF drugs used or on whether patients were anti-TNF naive or not.

### 2.2. Subset Definition

Subsets for gene ontology (GO) analysis were defined based on biomarker type. Preliminary subset analysis revealed no significant differences between the gene ontology terms of biomarkers measured in synovial fluid and those measured in sera. For this reason, we did not make any distinctions based on biomarker measurement locations.

Potential therapeutic targets can be either stimulated or blocked. In general, processes that are up-regulated in responders or down-regulated in non-responders could be stimulated to achieve better response or even restore response. Similarly, processes that are down-regulated in responders or up-regulated in non-responders can be blocked. Following this reasoning, we created two additional separate groups for RNA and protein biomarkers. The first group (_UP_R_DO_N) contains biomarkers reported either as up-regulated in responders or down-regulated in non-responders; the second group (_DO_R_UP_N) contain biomarkers down-regulated in responders or up-regulated in non-responders.

To enhance biological process discovery with gene ontology analysis, gene networks were constructed. In this study, “gene network” refers to a set of interacting biomarkers produced from a list of biomarkers of interest (i.e., previously published anti-TNF response biomarkers). Biomarkers interacting with at least two biomarkers of interest were obtained from BIOGRID [13,14] using the biogridR package [15] for R (version 4.1.1, R Core Team, Vienna, Austria) [16].

Subset names are defined in Table 1.

### 2.3. Gene Ontology Analysis

Gene ontology analysis was performed using the software package CytoScape (v3.8.2., CytoScape Team) [17] with the integrated application ClueGO (v2.5.8, Laboratory of Integrative Cancer Immunology (Team 15), Paris, France) [18]. ClueGO analysis was performed using the following parameters and selected options:Ontology/pathways selected:○Biological Process (13 May 2021);○Cellular Component (13 May 2021);○Molecular Function (13 May 2021);
Evidence selected: only *All_Experimental*.


Moreover, comparative gene ontology analysis was employed to estimate GO term specificity between different subsets (e.g., _UP_RE_DO_NR vs. _UP_NR_DO_RE).

Statistical significance was defined as a *p*-value lower than 5 × 10^−2^ after Bonferonni step-down correction (the default selection in ClueGO v2.5.8).

Gene ontology analysis results were visualized using default CytoScape settings and freely available style options.

## 3. Results

### 3.1. Literature Search

Using the defined search query (see Materials and Methods—Literature Search), we obtained 185 results in the PubMed database. Based on the inclusion criteria, 125 studies were included in the gene ontology analysis. Among the 125 studies, 61 studies reported DNA biomarkers, 15 studies reported RNA biomarkers, 39 studies reported protein biomarkers, while 10 studies reported response biomarkers that could not be categorized as DNA, RNA or protein biomarkers as they were cell counts, nuclear magnetic resonance (NMR) spectra or metabolomic markers. In addition, five studies reported biomarkers at several molecular levels.

Use of technologies to comprehensively study the genome, transcriptome and proteome remains uncommon, but it has become more common in recent years. Among the 61 DNA biomarker studies, 8 employed next-generation sequencing (NGS) technology and 3 out of 15 RNA biomarker studies employed RNA sequencing (RNAseq). Similarly, 7 out of 39 protein biomarker studies used liquid chromatography with mass spectrometry (LC–MS/MS) for biomarker discovery.

### 3.2. Biomarker Collection

The biomarkers extracted from the studies gathered from the literature are shown in Table 2 (DNA biomarkers), Table 3 (RNA biomarkers) and Table 4 (protein biomarkers). For gene ontology (GO) analysis, only biomarkers indexed in GO datasets can be processed. To remove potential duplicate biomarkers and obsolete gene names, we used the g:Convert Gene ID Converter tool [19] to update the biomarker names to the most recent ones. Finally, biomarkers that could not be reliably assigned to a gene with GO definitions were excluded (e.g., intergenic genetic variants).

Studies reporting biomarkers that could not be categorized as DNA, RNA or protein biomarkers are displayed below in Table 5.

### 3.3. Gene Ontology Analysis Results

The DNA subset has enriched GO terms related to the definition of non-response, while the DNA gene network only expanded upon the terms NF-κB signaling and TNF-α processes.

Gene ontology analysis of DNA biomarkers revealed terms already known to be associated with anti-TNF therapy non-response in rheumatoid arthritis, namely, terms connected to the definition of non-response or anti-TNF therapy, such as inflammation, tumor necrosis factor alpha, NF-κB signaling, IL-1, IL-2, IL-6 and IL-27. A subset of the terms related to NF-κB signaling is displayed in Figure 1.

RNA biomarker subsets revealed several enriched GO terms that were not previously directly associated with anti-TNF therapy response in rheumatoid arthritis. Such enriched terms in RNA subsets include prostaglandin synthesis, response to lipopolysaccharide (LPS), interferon gamma and macrophage chemotaxis. Gene networks based on RNA biomarkers and their BIOGRID interactors revealed novel significantly enriched GO terms related to the proteasome; the term *proteasome-mediated ubiquitin-dependent protein catabolic process* (*p* = 2.91 × 10^−15^) is a significant novel hyponym. The gene ontology terms related to the proteasome and others identified in the BIOGRID RNA biomarker network are illustrated in Figure 2.

Similarly, protein subsets also revealed several enriched GO terms that were not previously directly associated with anti-TNF therapy response in rheumatoid arthritis. Gene ontology analysis revealed several enriched blood lipoprotein (HDL, VLDL and cholesterol) terms, illustrated in Figure 3.

The full results of the gene ontology subset analysis are available in Appendix A.

BIOGRID data gene networks based on DNA and protein biomarkers did not reveal any novel enriched GO terms but expanded the associated hyponyms of leading GO terms. 

Comparative GO analysis of DNA, RNA and protein biomarkers showed no novel differences between analyzed subsets based on biomarker type. NF-κB signaling terms are specific to DNA, MHC protein complex terms are specific for RNA, while lipoprotein terms are specific to protein biomarkers. 

## 4. Discussion

The results of our study help to elucidate the mechanisms underlying response and non-response to anti-TNF therapy in rheumatoid arthritis. Biological markers linked to mechanisms associated with response and/or non-response to anti-TNF therapy have potential clinical applications as response predictors before or during anti-TNF therapy or even as potential novel therapeutic targets.

First, there was significant enrichment of protein metabolism terms in gene network subsets based on RNA biomarkers (specifically, RNA_UP_R_DO_N_BIO). The leading GO term was the hypernym *positive regulation of protein metabolic process* (*p* = 3.63 × 10^−37^). Specifically, several enriched hyponyms under this leading term are associated with the proteasome, such as *proteasome-mediated ubiquitin-dependent protein catabolic process* (*p* = 2.91 × 10^−15^). To our best knowledge, proteasome processes have not yet been implicated in anti-TNF therapy response in rheumatoid arthritis. In RA, the autophagy and proteasome protein degradation pathways are key processes for synovial fibroblast survival [141]. In response to TNFα, the autophagy pathway, but not the proteasome, is consistently stimulated, yet there is an increased dependence on the proteasome for cell viability [141]. If autophagy is blocked in the presence of TNFα, an increase in proteasome activity occurs in some RA synovial fibroblasts but decreases in healthy synovial fibroblasts [141]. Targeting the proteasome complex thus represents a therapeutic opportunity to decrease synovial fibroblast survival, pannus growth and inflammation in RA [142,143,144]. Bortezomib, a proteasome inhibitor indicated for hematological cancers, was shown to decrease bone loss in an animal model of RA [145] and inflammatory cytokine production in an ex vivo study of activated T cells of healthy controls and RA patients [146]. In a recent study, delanzomib, a novel proteasome inhibitor, was successfully used together with adalimumab in a rat model of rheumatoid arthritis [147]. Moreover, two case reports showed remission of rheumatoid arthritis complicated with multiple myeloma [148] or TEMPI syndrome [149] after administration of bortezomib. 

Second, several terms related to lipoproteins were found to be significantly enriched in protein biomarker subsets. In the subset containing all protein biomarkers, the leading lipoprotein terms were *lipoprotein particle receptor binding* (*p* = 8.81 × 10^−12^) and *plasma lipoprotein particle* (*p* = 4.55 × 10^−11^). Interestingly, the hyponyms *very-low-density lipoprotein particle* (*p* = 1.83 × 10^−10^) and *spherical high-density lipoprotein particle* (*p* = 5.22 × 10^−8^) suggest the role of very-low-density lipoproteins (VLDLs) and high-density lipoproteins (HDLs) in response. Comparative GO analysis showed VLDL to be specific for protein biomarkers down-regulated in responders (or up-regulated in non-responders), and HDL was shown to be up-regulated in responders (or down-regulated in non-responders). These findings confirm clinical observations of increased HDL [150,151] as well as triglyceride and total cholesterol levels [152] after anti-TNF therapy initiation. Moreover, low baseline VLDL has been linked with a better response to anti-TNF therapy [153], which coincides with our finding of VLDLs being down-regulated in responders. Although blood lipid profiles may only reflect systemic inflammation and thus also disease severity, their role in anti-TNF therapy response is not yet understood. Blood lipid profiles are potential accessible and affordable anti-TNF response biomarkers that could be integrated into clinical routine.

Third, our results show a significant enrichment of GO terms related to leukocyte chemotaxis in RNA subsets, with the leading term being *negative regulation of leukocyte chemotaxis* (*p* = 3.26 × 10^−4^). Hyponym investigation in a comparative analysis of RNA biomarkers up-regulated and down-regulated in responders showed the term *negative regulation of macrophage chemotaxis* (*p* = 3.00 × 10^−3^) to be up-regulated in responders (or down-regulated in non-responders). This finding suggests that good responders have lower macrophage infiltration than non-responders. Macrophage chemotaxis thus represents both an opportunity for response biomarker discovery as well as a therapeutic target. An example of a leukocyte chemotaxis reducing drug is montelukast, a cysteinyl leukotriene receptor antagonist used to treat asthma and allergic rhinitis. Although montelukast is mainly used to block leukotriene-dependent human airway smooth muscle contractions, it also blocks up-regulation of vascular permeability and leukocyte chemotaxis. A study has shown that montelukast decreases inflammatory cytokine production in RA and thus represents a novel therapeutic strategy [154].

Finally, our review of anti-TNF therapy response biomarkers has revealed that many response biomarkers have been reported at several levels of biological data (DNA, RNA, proteins, etc.), but only 12 biomarkers were reported by more than one study. Biomarkers reported by more than one study include the DNA biomarkers *CCL4* and *IL1B*; the RNA biomarkers *FCGR2A*, *FCGR3A*, *IL10*, *IL6*, *PTPRC* and *TNF*; and the protein biomarkers IL6, ITIH1, S100A8 and S100A9. Recently, a Japanese cohort has demonstrated the use of interferon signatures and their dynamics for use in long-term anti-TNF drug response prediction, which validates previously reported biomarkers related to interferon proteins [155]. Interestingly, results from another recent study showed that interferon-related chemokine levels (e.g., CXCL10) correlated with disease activity but not with short-term response to anti-TNF therapy (certolizumab pegol) in a Swedish cohort [156]. These studies highlight the difficulties of biomarker replication, especially with cohorts from different ethnic backgrounds and with different study designs.

Our GO analysis of anti-TNF therapy response biomarkers highlighted several biological processes as significantly enriched in response and/or non-response to anti-TNF therapy. Our results encourage targeted analysis of these biological processes for novel biomarker discovery but also the development of novel therapeutic strategies in the treatment of RA. The highlighted therapeutic targets could be useful either as alternatives for anti-TNF therapy non-responders, as co-therapies with anti-TNF treatment or as novel maintenance strategies. Moreover, our study’s review of anti-TNF response biomarkers revealed that although response biomarkers have been extensively studied, there is a generally low rate of overlap and biomarker validation between studies.

## 5. Conclusions

Biological processes related to the proteasome and blood lipids could affect response to anti-TNF therapy according to gene ontology of existing anti-TNF therapy response biomarkers in RA. Our study encourages further investigation of proteasome and blood lipid processes in RA anti-TNF response.

## Figures and Tables

**Figure 1 biomedicines-10-01808-f001:**
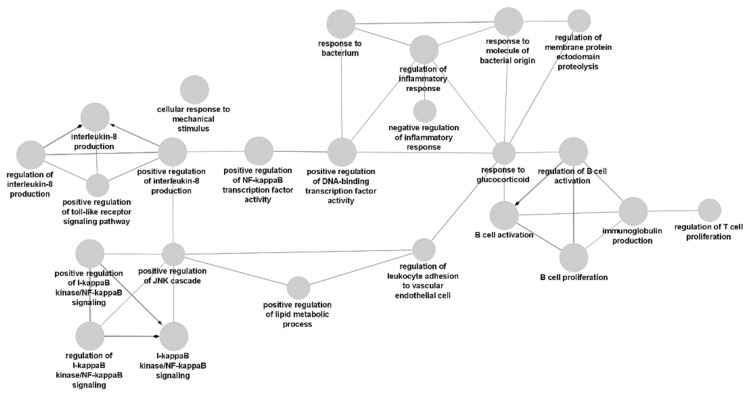
Extended network of gene ontology term nodes related to NF-κB signaling, as identified in the DNA biomarker subset.

**Figure 2 biomedicines-10-01808-f002:**
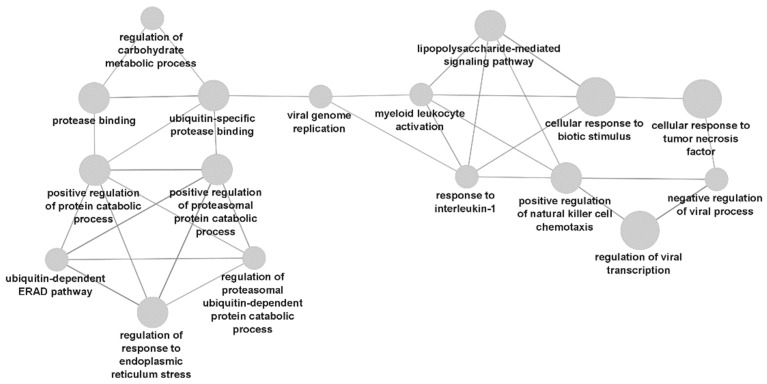
Network of gene ontology term nodes related to the proteasome, as identified in RNA biomarker subsets with BIOGRID data.

**Figure 3 biomedicines-10-01808-f003:**
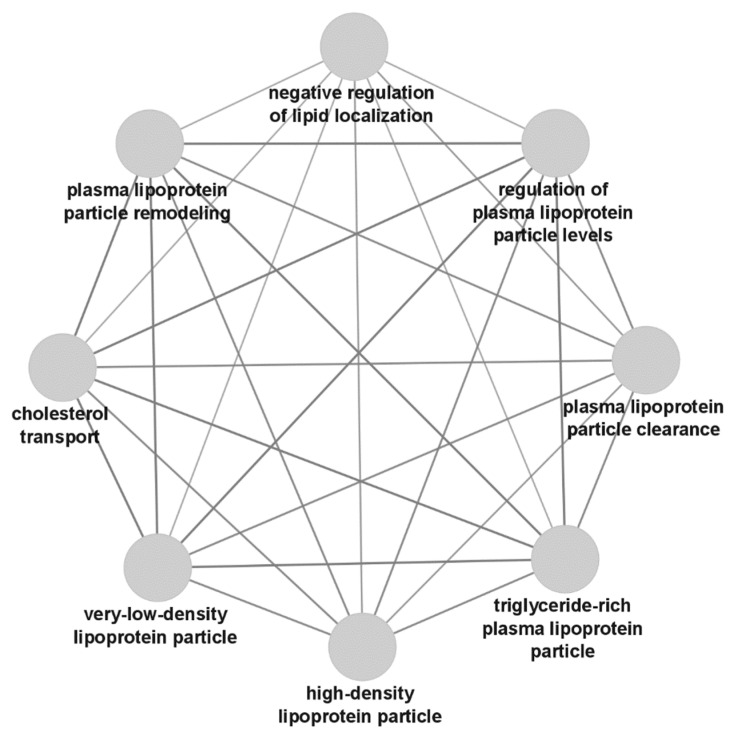
Extended network of gene ontology term nodes related to lipids, as identified in the protein biomarker subset.

**Table 1 biomedicines-10-01808-t001:** Biomarker subsets. Subset names are constructed using biomarker type (DNA, RNA or PRO for protein) followed by association type (_UP_R_DO_N or _DO_R_UP_N) and indicate whether or not a given subset is a gene network derived from BIOGRID data (_BIO).

Subset Name	Biomarkers Included in Subset
DNA	All DNA biomarkers
RNA	All RNA biomarkers
RNA_UP_R_DO_N	RNA biomarkers up-regulated in responders or down-regulated in non-responders
RNA_DO_R_UP_N	RNA biomarkers up-regulated in non-responders or down-regulated in responders
PRO	All protein biomarkers
PRO_UP_R_DO_N	Protein biomarkers up-regulated in responders or down-regulated in non-responders
PRO_DO_R_UP_N	Protein biomarkers up-regulated in non-responders or down-regulated in responders
DNA_BIO	BIOGRID network based on DNA biomarkers
RNA_BIO	BIOGRID network based on RNA biomarkers
RNA_UP_R_DO_N_BIO	BIOGRID network based on RNA biomarkers up-regulated in responders or down-regulated in non-responders
RNA_DO_R_UP_N_BIO	BIOGRID network based on RNA biomarkers up-regulated in non-responders or down-regulated in responders
PRO_BIO	BIOGRID network based on protein biomarkers
PRO_UP_R_DO_N_BIO	BIOGRID network based on protein biomarkers up-regulated in responders or down-regulated in non-responders
PRO_DO_R_UP_N_BIO	BIOGRID network based on protein biomarkers up-regulated in non-responders or down-regulated in responders

**Table 2 biomedicines-10-01808-t002:** DNA biomarkers of response to anti-TNF therapy in RA.

Study	Associated Gene
Criswell, L.A. et al., 2004 [20]	*TNF* *LTA* *HLA-DRB1*
Lee, Y.H. et al., 2006 [21]	*TNF*
Ongaro, A. et al., 2008 [22]	*TNFSFR1B*
Jančić, I. et al., 2013 [23]	*IL6*
Lee, Y.H. et al., 2014 [24]	*IL6*
Lee, Y.H. et al., 2016 [25]	*PTPRC* *FCGR2A*
Schotte, H. et al., 2015 [26]	*IL6*
Pappas, D.A. et al., 2013 [27]	*CCL21* *CD28*
Morales-Lara, M.J. et al., 2012 [28]	*TRAILR1* *TNFR1A*
Pers, Y.M. et al., 2014 [29]	*TNFSFR1B*
Iwaszko, M. et al., 2016 [30]	*KLRD1* *KLRC1*
O’Rielly, D.D. et al., 2009 [31]	*TNF*
Ferreiro-Iglesias, A. et al., 2016 [32]	*PTPRC* *IL10* *CHUK*
Julià, A. et al., 2016 [33]	*MED15*
Kang, C.P. et al., 2005 [34]	*TNF*
Seitz, M. et al., 2007 [35]	*TNF*
Iannaccone, C.K. et al., 2011 [36]	*PTPRC*
Dávila-Fajardo, C.L. et al., 2014 [37]	*IL6*
Montes, A. et al., 2014 [38]	*FCGR2A*
Bowes, J.D. et al., 2009 [39]	*MAP3K1* *MAP3K14*
Miceli-Richard, C. et al., 2008 [40]	*HLA-DRB1*
Tsukahara, S. et al., 2008 [41]	*FCGR3A*
Cañete, J.D. et al., 2009 [42]	*FCGR2A* *FCGR3A*
Potter, C. et al., 2010 [43]	*MYD88* *CHUK*
Coulthard, L.R. et al., 2011 [44]	*MAP2K6* *MSK1* *MSK2* *MAPK14*
Acosta-Colman, I. et al., 2013 [45]	*PDE3A*
Dávila-Fajardo, C.L. et al., 2015 [46]	*FCGR2A*
Sun, Y. et al., 2017 [47]	*FCGR2A* *FCGR3A*
Morales-Lara, M.J. et al., 2010 [48]	*FCGR3A*
Lee, Y.H. et al., 2010 [49]	*TNF*
Liu, C. et al., 2008 [50]	*LMO4* *GBP6* *CERS6* *ARAP2* *QKI* *PON1* *IFNK* *MOB3B* *C9orf72* *MAFB* *CST5*
Tan, R.J. et al., 2010 [51]	*AFF3* *CD226*
Plant, D. et al., 2011 [52]	*EYA4* *PDZD2*
McGeough, C.M. et al., 2012 [53]	*HLA-C*
Krintel, S.B. et al., 2012 [54]	*CD19* *STXBP6*
Plant, D. et al., 2012 [55]	*PTPRC*
Cui, J. et al., 2013 [56]	*CD84*
Cui, J. et al., 2010 [57]	*PTPRC*
Sode, J. et al., 2014 [58]	*NLRP3*
Umiċeviċ Mirkov, M. et al., 2013 [59]	*CNTN5* *NUBPL*
Canhão, H. et al., 2015 [60]	*TRAF1*
Avila-Pedretti, G. et al., 2015 [61]	*FCGR2A*
Schotte, H. et al., 2015 [62]	*IL10*
Sode, J. et al., 2015 [63]	*TLR1* *TLR5* *NLRP3*
Honne, K. et al., 2016 [64]	*MAP3K7* *BACH2* *WDR27* *GFRA1*
Jančić, I. et al., 2015 [65]	*TNF* *IL6*
Folkersen, L. et al., 2016 [66]	*MAFB*
Gębura, K. et al., 2017 [67]	*TLR9* *NFKB1*
Nishimoto, T. et al., 2014 [68]	*TRAF1*
Sarsour, K. et al., 2013 [69]	*FCGR3A*
Vasilopoulos, Y. et al., 2011 [70]	*TNFRSF1B* *TNF* *TNFRSF1A*
Rooryck, C. et al., 2008 [71]	*TNFRSF1B*
Cuchacovich, M. et al., 2006 [72]	*TNF*
Tutuncu, Z. et al., 2005 [73]	*FCGR3A*
Sode, J. et al., 2018 [74]	*IRAK3* *CHUK* *MYD88* *NFKBIB* *NLRP3*
Iwaszko, M. et al., 2018 [75]	*NKG2D*
Skapenko, A. et al., 2019 [76]	*HLA-DRB1* *IL4R* *FCGR2B*
Spiliopoulou, A. et al., 2019 [77]	*CD40* *ENTPD1*
Wielińska, J. et al., 2020 [78]	*RANK* *RANKL*
Gibson, D.S. et al., 2021 [79]	*CD226* *HLA-DRB1*
Iwaszko, M. et al., 2021 [80]	*IL33*

**Table 3 biomedicines-10-01808-t003:** RNA biomarkers of response to anti-TNF therapy in RA.

Study	Gene	Association Direction
Stuhlmüller, B. et al., 2010 [81]	*CD11C*	Up-regulated in responders
Sekiguchi, N. et al., 2008 [82]	*HLA-DQA1*	Down-regulated in non-responders
	*IGHM*	Down-regulated in non-responders
	*AP1S2*	Up-regulated in non-responders
Wright, H.L. et al., 2015 [83]	*IFNG*	Up-regulated in responders
Wright, H.L. et al., 2016 [84]	*CMPK2*	Up-regulated in responders
	*IFIT1B*	Up-regulated in responders
	*RNASE3*	Up-regulated in responders
Tsuzaka, K. et al., 2010 [85]	*ADAMTS5*	Down-regulated in responders
Oliveira, R.D. et al., 2012 [86]	*CCL4*	Up-regulated in responders
	*CD83*	Up-regulated in responders
	*BCL2A1*	Up-regulated in responders
Lequerré, T. et al., 2006 [87]	*CYP3A4*	Down-regulated in responders
	*AKAP9*	Down-regulated in responders
	*LAMR1*	Down-regulated in responders
	*FBXO5*	Down-regulated in responders
	*RASGRP3*	Down-regulated in responders
	*PFKFB4*	Down-regulated in responders
	*HLA-DPB1*	Down-regulated in responders
	*PSMB9*	Down-regulated in responders
	*EPS15*	Down-regulated in responders
	*MTCBP-1*	Down-regulated in responders
	*MRPL22*	Up-regulated in responders
	*MCP*	Up-regulated in responders
	*KNG1*	Up-regulated in responders
	*AADAT*	Up-regulated in responders
Koczan, D. et al., 2008 [88]	*TNFAIP3*	Down-regulated in responders
	*NFKBIA*	Down-regulated in responders
	*RUNX1*	Up-regulated in responders
	*ZFP36L2*	Down-regulated in responders
	*IL1B*	Down-regulated in responders
	*IL1B*	Down-regulated in responders
	*CCL4*	Down-regulated in responders
	*CCL3*	Down-regulated in responders
	*CXCL2*	Down-regulated in responders
	*ADAM12*	Down-regulated in responders
	*SCN2B*	Up-regulated in responders
	*PDE4B*	Down-regulated in responders
	*RAPGEF1*	Down-regulated in responders
	*MYO10*	Down-regulated in responders
	*PTPRD*	Up-regulated in responders
	*PDE4B*	Down-regulated in responders
	*LGALS13*	Up-regulated in responders
	*CHST3*	Down-regulated in responders
	*LUC7L3*	Up-regulated in responders
	*PPP1R15A*	Down-regulated in responders
	*ADM*	Down-regulated in responders
	*CHRND*	Down-regulated in responders
	*PIGO*	Down-regulated in responders
	*RNF19B*	Down-regulated in responders
	*FSD1*	Down-regulated in responders
van Baarsen, L.G. et al., 2010 [89]	*OAS1*	Up-regulated in non-responders
	*LGALS3BP*	Up-regulated in non-responders
	*MX2*	Up-regulated in non-responders
	*OAS2*	Up-regulated in non-responders
	*SERPING1*	Up-regulated in non-responders
Toonen, E.J. et al., 2012 [90]	*HIRIP3*	Down-regulated in responders
	*TPM1*	Up-regulated in responders
	*NPRL2*	Down-regulated in responders
	*CLIC3*	Down-regulated in responders
	*PTGS2*	Up-regulated in responders
	*G0S2*	Up-regulated in responders
	*PIGV*	Down-regulated in responders
	*HIF1A*	Up-regulated in responders
	*ZBTB6*	Down-regulated in responders
	*RANBP17*	Up-regulated in responders
	*PCGF5*	Up-regulated in responders
	*SESTD1*	Up-regulated in responders
	*GPD2*	Up-regulated in responders
	*HERPUD2*	Up-regulated in responders
	*DND1*	Down-regulated in responders
	*SH2D2A*	Down-regulated in responders
	*EIF4E2*	Down-regulated in responders
	*GTPBP2*	Up-regulated in responders
	*TPRA1*	Down-regulated in responders
	*GRAMD1B*	Up-regulated in responders
	*PPP1R15A*	Up-regulated in responders
	*PMAIP1*	Up-regulated in responders
	*RAPGEF1*	Up-regulated in responders
	*CSRNP1*	Up-regulated in responders
	*TMOD2*	Up-regulated in responders
	*EGR2*	Up-regulated in responders
	*DUSP1*	Up-regulated in responders
	*MTURN*	Up-regulated in responders
	*EGR3*	Up-regulated in responders
	*SQSTM1*	Up-regulated in responders
	*RAMP3*	Down-regulated in responders
	*PDE3A*	Up-regulated in responders
	*VEPH1*	Up-regulated in responders
	*GBP7*	Up-regulated in responders
	*PSTPIP2*	Up-regulated in responders
	*FAM221A*	Down-regulated in responders
	*ZNF2*	Down-regulated in responders
	*MED12L*	Up-regulated in responders
	*OSM*	Down-regulated in responders
	*TMEM186*	Down-regulated in responders
	*PKHD1L1*	Up-regulated in responders
	*OR6C74*	Down-regulated in responders
	*GPN2*	Down-regulated in responders
	*DDX39B*	Down-regulated in responders
	*UNQ5840*	Down-regulated in responders
	*C15ORF40*	Down-regulated in responders
	*CMIP*	Up-regulated in responders
	*KCNJ13*	Down-regulated in responders
	*SLC7A6OS*	Down-regulated in responders
	*ELOVL4*	Down-regulated in responders
	*UQCRFS1*	Down-regulated in responders
	*NBN*	Up-regulated in responders
	*BEX2*	Down-regulated in responders
	*YPEL5*	Up-regulated in responders
	*FAIM*	Down-regulated in responders
	*STAT1*	Up-regulated in responders
	*CXCL8*	Down-regulated in responders
	*PIH1D2*	Down-regulated in responders
	*EDC3*	Down-regulated in responders
	*TNFAIP3*	Up-regulated in responders
	*FSCN1*	Down-regulated in responders
	*MGLL*	Up-regulated in responders
	*GCNT2*	Up-regulated in responders
	*EGF*	Up-regulated in responders
	*COLGALT2*	Down-regulated in responders
	*HOPX*	Down-regulated in responders
	*NT5C3A*	Up-regulated in responders
	*RNF11*	Up-regulated in responders
	*SLK*	Up-regulated in responders
	*TAP2*	Up-regulated in responders
	*GBP1*	Up-regulated in responders
	*GBP5*	Up-regulated in responders
	*XRN1*	Up-regulated in responders
	*PTGDS*	Down-regulated in responders
	*TAS2R50*	Up-regulated in responders
	*HSPC159*	Up-regulated in responders
	*ARL6*	Down-regulated in responders
	*PDE4B*	Up-regulated in responders
	*OR2L3*	Down-regulated in responders
	*NR4A2*	Up-regulated in responders
	*PALD1*	Down-regulated in responders
	*OGG1*	Down-regulated in responders
	*ADGRE5*	Up-regulated in responders
	*FRMD3*	Up-regulated in responders
	*LRRIQ3*	Down-regulated in responders
	*RAD23A*	Down-regulated in responders
	*APP*	Up-regulated in responders
	*PXT1*	Down-regulated in responders
	*MPP7*	Up-regulated in responders
	*NEXN*	Up-regulated in responders
	*GMPR*	Up-regulated in responders
	*UVRAG*	Up-regulated in responders
	*ADAMTS1*	Down-regulated in responders
	*ATP6V0A2*	Down-regulated in responders
	*CATSPER3*	Down-regulated in responders
	*C5*	Up-regulated in responders
	*MAP4K2*	Up-regulated in responders
	*GCH1*	Up-regulated in responders
	*ATP6V0E2*	Down-regulated in responders
	*FBXO10*	Down-regulated in responders
	*ZNF425*	Down-regulated in responders
	*HSCB*	Down-regulated in responders
	*GTF2F2*	Up-regulated in responders
	*PGK1*	Down-regulated in responders
	*STAT2*	Up-regulated in responders
	*PCSK6*	Up-regulated in responders
	*TMEM268*	Up-regulated in responders
	*PPCDC*	Up-regulated in responders
	*GSX1*	Down-regulated in responders
Cui, J. et al., 2013 [56]	*CD84*	Up-regulated in responders
Thomson, T.M. et al., 2015 [91]	*FOXA2*	Up-regulated in non-responders
	*ERBB2*	Up-regulated in non-responders
	*IL11*	Up-regulated in non-responders
	*MAP2K3*	Up-regulated in non-responders
	*NF1*	Down-regulated in non-responders
	*S100A9*	Down-regulated in non-responders
	*S100A8*	Down-regulated in non-responders
	*MST1R*	Down-regulated in non-responders
	*NOS2*	Down-regulated in non-responders
	*NR2F6*	Down-regulated in non-responders
	*PPARG*	Up-regulated in non-responders
	*MEIS1*	Up-regulated in non-responders
	*DPPA4*	Up-regulated in non-responders
	*MBD1*	Down-regulated in non-responders
	*CDK2*	Up-regulated in non-responders
Folkersen, L. et al., 2016 [66]	*SORBS3*	Down-regulated in responders
	*AKAP9*	Down-regulated in responders
Póliska, S. et al., 2019 [92]	*TMEM176A*	Up-regulated in responders
	*TMEM176B*	Up-regulated in responders
	*PLSCR1*	Up-regulated in responders
	*IFI44*	Up-regulated in responders
Oliver, J. et al., 2021 [93]	*LIN7A*	Down-regulated in responders
	*CREB5*	Down-regulated in responders
	*ENTPD1*	Down-regulated in responders
	*ITGB7*	Up-regulated in responders
	*HLA-DMA*	Up-regulated in responders
	*IL6R*	Down-regulated in responders
	*SLC8A1*	Down-regulated in responders
	*IL1B*	Down-regulated in responders
	*HLA-DOB*	Up-regulated in responders
	*MGAM*	Down-regulated in responders
	*TRAF5*	Up-regulated in responders
	*AES*	Up-regulated in responders
	*E2F5*	Up-regulated in responders
	*ZFYVE16*	Down-regulated in responders
	*HLA-DOA*	Up-regulated in responders
	*TLR8*	Down-regulated in responders
	*STAP1*	Up-regulated in responders
	*TGM3*	Down-regulated in responders
	*PI3*	Down-regulated in responders
	*ARG1*	Down-regulated in responders
	*MMP9*	Down-regulated in responders
	*MGAM*	Down-regulated in responders
	*CA4*	Down-regulated in responders
	*KAZN*	Down-regulated in responders
	*PGLYRP1*	Down-regulated in responders
	*FCAR*	Down-regulated in responders
	*PROK2*	Down-regulated in responders
	*MANSC1*	Down-regulated in responders
	*TRPM6*	Down-regulated in responders
	*SLC26A8*	Down-regulated in responders
	*SULT1B1*	Down-regulated in responders
	*IL1R1*	Down-regulated in responders
	*MAK*	Down-regulated in responders
	*ADM*	Down-regulated in responders
	*TMEM88*	Down-regulated in responders
	*CYP4F3*	Down-regulated in responders
	*REPS2*	Down-regulated in responders
	*ANXA3*	Down-regulated in responders
	*ABCA1*	Down-regulated in responders
	*F5*	Down-regulated in responders
	*ANPEP*	Down-regulated in responders
	*EPSTI1*	Up-regulated in responders
	*SERPING1*	Up-regulated in responders
	*MS4A1*	Up-regulated in responders
	*C1QA*	Up-regulated in responders
	*BATF2*	Up-regulated in responders
	*FCRLA*	Up-regulated in responders
	*IGLL5*	Up-regulated in responders
	*MZB1*	Up-regulated in responders
	*IGJ*	Up-regulated in responders

**Table 4 biomedicines-10-01808-t004:** Protein biomarkers of response to anti-TNF therapy in RA.

Study	Protein Marker	Association Direction
Straub, R.H. et al., 2008 [94]	Cortisol	Down-regulated in responders
Ammitzbøll, C.G. et al., 2013 [95]	FCN1	Down-regulated in responders
Matsuyama, Y. et al., 2012 [96]	IL33	Down-regulated in responders
	IL33	Down-regulated in responders
Morozzi, G. et al., 2007 [97]	COMP	Down-regulated in responders
Kohno, M. et al., 2008 [98]	IL17 to TNF ratio	Down-regulated in responders
Ortea, I. et al., 2012 [99]	GC	Up-regulated in non-responders
	CP	Up-regulated in non-responders
	APOB	Up-regulated in non-responders
	ITIH2	Up-regulated in non-responders
	THBS1	Up-regulated in non-responders
	C4B	Up-regulated in non-responders
	ITIH1	Up-regulated in non-responders
	GSN	Up-regulated in non-responders
	APOA2	Up-regulated in non-responders
	FN1	Up-regulated in non-responders
	CFHR4	Up-regulated in non-responders
	APOM	Up-regulated in non-responders
	APMAP	Up-regulated in non-responders
	MASP2	Up-regulated in non-responders
Shi, R. et al., 2018 [100]	BIRC5	Down-regulated in responders
	CRP	Up-regulated in responders
	IL6	Up-regulated in responders
Cañete, J.D. et al., 2011 [101]	TNFRSF1B	Up-regulated in responders
Kayakabe, K. et al., 2012 [102]	IL1B	Down-regulated in non-responders
Sakthiswary, R. et al., 2014 [103]	IgA rheumatoid factor	Up-regulated in non-responders
Andersen, M. et al., 2017 [104]	MC1R	Down-regulated in responders
	MC3R	Down-regulated in responders
	MC5R	Down-regulated in responders
	MC1R	Down-regulated in responders
	MC3R	Down-regulated in responders
	MC5R	Down-regulated in responders
Choi, I.Y. et al., 2015 [105]	S100A8/S100A9 complex	Up-regulated in responders
La, D.T. et al., 2008 [106]	TNFSF13B	Down-regulated in responders
Odai, T. et al., 2009 [107]	CX3CL1	Down-regulated in responders
Kuuliala, A. et al., 2006 [108]	IL2	Down-regulated in responders
González-Alvaro, I. et al., 2007 [109]	TNFSF11	Down-regulated in responders
Fabre, S. et al., 2008 [110]	CCL2	Down-regulated in non-responders
	EGF	Down-regulated in non-responders
Wijbrandts, C.A. et al., 2008 [111]	TNF	Up-regulated in responders
Hueber, W. et al., 2009 [112]	CSF2	Up-regulated in responders
	IL6	Up-regulated in responders
	FMOD	Up-regulated in responders
	CLU	Up-regulated in responders
	APOE	Up-regulated in responders
	HIST1H2BM	Up-regulated in responders
	HSP58	Up-regulated in responders
	IL1A	Up-regulated in responders
	COMP	Up-regulated in responders
	CAST	Up-regulated in responders
	BGN	Up-regulated in responders
	OGN	Up-regulated in responders
	TMPRSS11A	Up-regulated in responders
	IL1B	Up-regulated in responders
	CCL11	Up-regulated in responders
	CXCL10	Up-regulated in responders
	FGF1	Up-regulated in responders
	CCL2	Up-regulated in responders
	IL12P70	Up-regulated in responders
	IL12P40	Up-regulated in responders
	IL15	Up-regulated in responders
Lindberg, J. et al., 2010 [113]	LGALS1	Up-regulated in responders
	SCNN1B	Down-regulated in responders
	GMNN	Down-regulated in responders
	PALLD	Down-regulated in responders
	TPPP3	Up-regulated in responders
	LGALS1	Down-regulated in responders
	NONO	Down-regulated in responders
	ATP5H	Down-regulated in responders
	PGLS	Down-regulated in responders
	UBA52	Down-regulated in responders
	RPS12	Down-regulated in responders
	RPLP0P6	Down-regulated in responders
	ANAPC11	Down-regulated in responders
	PGA3	Up-regulated in responders
	WDR83OS	Down-regulated in responders
	MYO15A	Down-regulated in responders
	MRPL33	Down-regulated in responders
	FOXC2	Down-regulated in responders
	H3F3A	Down-regulated in responders
	FAP	Down-regulated in responders
	TRAF3IP2	Down-regulated in responders
	AGPAT4	Down-regulated in responders
	RPL36A	Up-regulated in responders
	RIN2	Down-regulated in responders
	RPL13A	Down-regulated in responders
	NEK5	Down-regulated in responders
	RPL7	Down-regulated in responders
Trocmé, C. et al., 2009 [114]	APOA1	Up-regulated in responders
	PF4	Up-regulated in non-responders
Chen, D.Y. et al., 2011 [115]	IL17	Up-regulated in non-responders
Meusch, U. et al., 2013 [116]	IL1R2	Up-regulated in responders
Obry, A. et al., 2014 [117]	S100A8	Up-regulated in responders
	S100A9	Up-regulated in responders
Blaschke, S. et al., 2015 [118]	Haptoglobin-α1	Up-regulated in responders
	Haptoglobin-α2	Up-regulated in responders
	HP	Up-regulated in responders
	GC	Up-regulated in responders
	APOC3	Up-regulated in non-responders
Zhang, F. et al., 2015 [119]	IL34	Down-regulated in responders
Meusch, U. et al., 2015 [120]	TNFRSF1A	Up-regulated in responders
	IL1RA	Up-regulated in responders
Obry, A. et al., 2015 [121]	STUB1	Up-regulated in responders
	PROS1	Up-regulated in responders
	C1R	Up-regulated in responders
	CPN2	Up-regulated in responders
	CP	Up-regulated in responders
	ITIH1	Up-regulated in responders
	ITIH3	Up-regulated in responders
	DYNC1I1	Up-regulated in responders
	S100A9	Up-regulated in responders
	AZGP1	Up-regulated in responders
	TF	Down-regulated in responders
	PLG	Up-regulated in responders
Nair, S.C. et al., 2016 [122]	S100A8–S100A9 complex	Up-regulated in responders
Ortea, I. et al., 2016 [123]	ADAMTSL2	Up-regulated in non-responders
	A2M	Up-regulated in non-responders
	APOA1	Down-regulated in non-responders
	APOA2	Up-regulated in non-responders
	APOB	Up-regulated in non-responders
	APOC1	Up-regulated in non-responders
	APOC3	Up-regulated in non-responders
	APOM	Up-regulated in non-responders
	F9	Up-regulated in non-responders
	CFL1	Up-regulated in non-responders
	C3	Up-regulated in non-responders
	C4B	Up-regulated in non-responders
	C8A	Up-regulated in non-responders
	CFHR4	Down-regulated in non-responders
	LGALS3BP	Up-regulated in non-responders
	HPX	Up-regulated in non-responders
	ITIH1	Up-regulated in non-responders
	ITIH2	Up-regulated in non-responders
	TPM3	Up-regulated in non-responders
	FN1	Up-regulated in non-responders
	MASP2	Up-regulated in non-responders
	PF4	Up-regulated in non-responders
	SH3BGRL3	Up-regulated in non-responders
	ABI3BP	Down-regulated in non-responders
	TCFL5	Down-regulated in non-responders
	TPM4	Up-regulated in non-responders
	TAGLN2	Up-regulated in non-responders
Wampler Muskardin, T. et al., 2016 [124]	IFN-β–α activity ratio	Up-regulated in non-responders
Folkersen, L. et al., 2016 [66]	ICAM1	Down-regulated in responders
	CXCL13	Up-regulated in responders
Nishimoto, T. et al., 2014 [68]	TRAF1	Up-regulated in non-responders
Koga, T. et al., 2011 [125]	PLAU	Up-regulated in responders
		Down-regulated in non-responders
Gerli, R. et al., 2008 [126]	CD30	Up-regulated in responders
Braun-Moscovici, Y. et al., 2006 [127]	IL6	Down-regulated in responders
Nguyen, M.V.C. et al., 2018 [128]	S100A12	Down-regulated in responders
	TTR	Up-regulated in responders
	PF4	Up-regulated in responders
Otsubo, H. et al., 2018 [129]	FOLR2	Up-regulated in non-responders
Frostegård, J. et al., 2021 [130]	PCSK9	Down-regulated in responders

**Table 5 biomedicines-10-01808-t005:** Markers which count not be categorized as DNA, RNA or protein biomarkers.

Study	Marker	Association Direction
Citro, A. et al., 2015 [131]	CD8+ T cells	Up-regulated in responders
Hull, D.N. et al., 2016 [132]	Th17 cells	Up-regulated in non-responders
Plant, D. et al., 2016 [133]	cg04857395	Down-regulated in responders
	cg26401028	Down-regulated in responders
	cg16426293	Down-regulated in responders
	cg03277049	Down-regulated in responders
	cg12226028	Down-regulated in responders
Talotta, R. et al., 2015 [134]	Th17 cells	Up-regulated in non-responders
	Th1 cells	Up-regulated in non-responders
Cuppen, B.V. et al., 2016 [135]	sn1-LPC (18:3-ω3/ω6)	Down-regulated in responders
	sn1-LPC (15:0)	Up-regulated in responders
	ethanolamine	Down-regulated in responders
	lysine	Up-regulated in responders
Chara, L. et al., 2012 [136]	CD14^+^highCD16^−^	Up-regulated in non-responders
	CD14^+^highCD16^+^	Up-regulated in non-responders
	CD14^+^lowCD16^+^	Up-regulated in non-responders
Alzabin, S. et al., 2012 [137]	Th17 cells	Up-regulated in non-responders
Klaasen, R. et. al., 2009 [138]	lymphocyte aggregates	Up-regulated in responders
Talotta, R. et al., 2016 [139]	Macrophages	Up-regulated in responders
Priori, R. et al., 2015 [140]	NMR spectra	Responder/non-responder specific

## Data Availability

Data are contained within the article or the Appendix A.

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
