# Peer review of "Gene Ontology Analysis Highlights Biological Processes Influencing Non-Response to Anti-TNF Therapy in Rheumatoid Arthritis"

_biomedicines, 2022, doi:10.3390/biomedicines10081808_

Round 1
Reviewer 1 Report
The manuscript entitled: Gene ontology analysis highlights biological processes influencing non-response to anti-TNF therapy in rheumatoid arthritis documents systematic elucidation of the biological processes underlying non-response to anti-TNF therapy in rheumatoid arthritis using the gene ontology. The manuscript is exciting. Below are my comments which may be useful to improve the quality of the manuscript.
Overall, the manuscript is comprehensive. In a few lines, the authors should discuss the other clinical interventions (and their limitations) for RA.
In 2022, several studies related to anti-TNF therapy in RA are documented. Including some of the relevant studies in the introduction or discussion will be helpful.
Authors should include the table with the ten studies under the title "Uncategorised" or something similar.
Line #34 Add a reference
Line 37# Cite the appropriate reference for each study
Figures #2 and #3 should be shuffled. Putting figures in the same order as the table (DNA biomarkers> RNA biomarkers> Protein biomarkers) will be helpful.
The manuscript has many abbreviations that should be described at least once in the text or authors can add a separate section of abbreviations at the end.
Authors should consider removing the "No of participants" column as it is not vital information. Instead, including the fold change of the RNA and protein biomarkers will make Tables more informative.
Minor:
The author's affiliation is not readable.
Author Response
Overall, the manuscript is comprehensive. In a few lines, the authors should discuss the other clinical interventions (and their limitations) for RA.
The following text has been added to the introduction to shortly address other pharmaceutical interventions and their limitations in RA:
“Patients who fail to respond to anti-TNF drugs may switch to a different biological drug, such as anakinra, rituximab or sarilumab [9]. Even so, other biological drugs face similar challenges with non-response as anti-TNF drugs [1,10,11]. Therefore, disease-modifying antirheumatic drugs (DMARDs) remain the long-term therapy of choice alongside corticosteroids for disease flares, both of which are known to have significant long term ad-verse effect [12].”
In 2022, several studies related to anti-TNF therapy in RA are documented. Including some of the relevant studies in the introduction or discussion will be helpful.
We added the following relevant studies published in 2022 to the document:
- Cacciapaglia, F. et al. Comparison of Adalimumab to Other Targeted Therapies in Rheumatoid Arthritis: Results from Systematic Literature Review and Meta-Analysis. J Pers Med 2022, 12, doi:10.3390/jpm1203035
- Iwasaki, T. et al., Dynamics of Type I and Type II Interferon Signature Determines Responsiveness to Anti-TNF Therapy in Rheumatoid Arthritis. Front Immunol 2022, 13, 901437, doi:10.3389/fimmu.2022.901437.
- Aldridge, J. et al. Blood chemokine levels are markers of disease activity but not predictors of remission in early rheumatoid arthritis. Clin Exp Rheumatol 2022, 40, 1393-1402, doi:10.55563/clinexprheumatol/idogmj.
Cacciapaglia, F. et al. is a recent comprehensive systematic review of adalimumab efficacy compared to other targeted therapies, covering more than a decade of observations.
Iwasaki, T. et al. and Aldridge, J. et. al. both report biomarkers related to interferon responses, but with seemingly conflicting results due to diverging study designs and different ethnic backgrounds of the analyzed cohorts.
Authors should include the table with the ten studies under the title "Uncategorised" or something similar.
The ten studies, including the markers these studies reported, are nowdisplayed in Table 5 titled “Markers which could not be categorized as DNA, RNA or protein biomarkers” directly below Table 4.
Line #34 Add a reference
A reference for the most recent and most extensive systematic review and meta-analysis of biological drug efficacy in rheumatoid arthritis has been added to line 34:
Cacciapaglia, F. et al. Comparison of Adalimumab to Other Targeted Therapies in Rheumatoid Arthritis: Results from Systematic Literature Review and Meta-Analysis. J Pers Med 2022, 12, doi:10.3390/jpm1203035
Line 37# Cite the appropriate reference for each study
Each biological drug has been cited appropriately with their originator study or early clinical trial studies. Line #35 has also been updated to better reflect the actual development of anti-TNF biological drugs. Specifically, etanercept is no longer mentioned as the first anti-TNF drug developed for rheumatoid arthritis since infliximab was developed under the name cA2 at the same time as etanercept.
Figures #2 and #3 should be shuffled. Putting figures in the same order as the table (DNA biomarkers> RNA biomarkers> Protein biomarkers) will be helpful.
The current order of figures follows the analysis type first (simple GO > GO with BIOGRID interacting marker nodes) and then the marker type (DNA > RNA > protein).
Nevertheless, we shuffled the figure order and changed the surrounding text under “Results” to accommodate the shuffled figures order.
The manuscript has many abbreviations that should be described at least once in the text or authors can add a separate section of abbreviations at the end.
Following descriptions of abbreviations have been added to the text:
- disease-modifying antirheumatic drugs (DMARDs)
- inhibitors of proinflammatory cytokine tumor necrosis factor alpha (anti-TNF)
- Disease Activity Score of 28 joints (ΔDAS28)
- gene ontology (GO)
- nuclear magnetic resonance (NMR)
- next generation sequencing (NGS)
- RNA sequencing (RNAseq)
- liquid chromatography with mass spectrometry (LC-MS/MS)
- lipopolysaccharide (LPS)
Authors should consider removing the "No of participants" column as it is not vital information. Instead, including the fold change of the RNA and protein biomarkers will make Tables more informative.
The column “No of participants” has been removed in Tables 2, 3 and 4.
We thank the reviewer for the suggestion to add the fold change to RNA and protein markers. However, given that this study is not a meta-analysis but a gene ontology analysis of subsets of biological markers, fold change values have no impact on the results for the studyand were thus not collected during the literature search and data extraction. Moreover, the direction of the association is already documented under the column “Association direction” and we believe fold changes would not add significantly more information.
Minor:
The author's affiliation is not readable.
The author’s affiliation font size is the Biomedicines MDPI format standard.
Reviewer 2 Report
Response to current standard therapy is a very important issue. However, we have very little knowledge about why some respond while some didn't. The authors utilize gene ontology to reveal the basic mechanism about treatment response and offer some new direction of further research.
The study dealt with biologic mechanisms under anti-TNF therapy. The study tried to answer this question through Gene ontology analysis. In my view, it was very interesting for a clinical physician such as me.
To my knowledge, utilizing gene ontology to point to a new research direction is a very novel approach and I didn't know of similar studies in this field. In this article, the author highlighted proteasome and lipoprotein pathways, which was a very interesting finding.
The paper is well written, the text is clear and easy to read, and the conclusions are consistent with the evidence provided and address the main question.
Author Response
We would like to kindly thank the reviewer for the comments.
This manuscript is a resubmission of an earlier submission. The following is a list of the peer review reports and author responses from that submission.